# Association between Bone Mineral Density and Bone Turnover Markers in Breast Cancer Patients and Bone-Only Metastasis

**DOI:** 10.3390/medicina57090880

**Published:** 2021-08-26

**Authors:** Berrin Papila Kundaktepe, Volkan Sozer, Fatih Orkun Kundaktepe, Sinem Durmus, Cigdem Papila, Hafize Uzun, Gonul Simsek, Remise Gelisgen

**Affiliations:** 1Department of General Surgery, Faculty of Cerrahpasa Medicine, Istanbul University—Cerrahpasa, 34098 Istanbul, Turkey; papilaberrin@yahoo.com; 2Department of Biochemistry, Yildiz Technical University, 34220 Istanbul, Turkey; volkansozer2000@yahoo.com; 3Department of Internal Medicine, Gaziosmanpaşa Taksim Research and Training Hospital, 34433 Istanbul, Turkey; doktorkun@gmail.com; 4Department of Medical Biochemistry, Faculty of Cerrahpasa Medicine, Istanbul University—Cerrahpasa, 34098 Istanbul, Turkey; durmus.sinem@gmail.com (S.D.); remisagelisgen@hotmail.com (R.G.); 5Department of Internal Medicine, Oncology Division, Faculty of Cerrahpasa Medicine, Istanbul University—Cerrahpasa, 34098 Istanbul, Turkey; ipapila56@yahoo.com; 6Department of Physiology, Faculty of Cerrahpasa Medicine, Istanbul University—Cerrahpasa, 34098 Istanbul, Turkey; gdincsimsek@yahoo.com

**Keywords:** breast cancer, bone mineral density, bone turnover markers, N-terminal cross-linking telopeptide, C-terminal cross-linking telopeptide of type I collagen

## Abstract

*Background and Objectives*: The aim of this study was to determine the influence of bone turnover markers, namely the N-terminal cross-linking telopeptide (NTx) and alpha C-terminal cross-linking telopeptide of type I collagen (α-CTx), in detecting bone metastasis (bone-only) in breast cancer (BC) patients, as well as to determine whether this effect is related to changes in bone mineral density (BMD). *Materials and Methods*: The participants in this study comprised 30 postmenopausal BC patients with bone metastases (age range: 59.56 ± 9.02), 20 postmenopausal BC patients without bone metastases (age range: 55.30 ± 11.55), and 20 healthy postmenopausal female controls (age range: 55.55 ± 5.85). Bone turnover markers (serum NTx and urine α-CTx) were measured using the ELISA method. A densitometer using dual-energy X-ray absorptiometry (DEXA) was used to analyze the BMD, and tumor markers were measured using the chemiluminescent immunometric assay. *Results*: The corresponding levels of serum NTx (*p* = 0.004), parathyroid hormone (PTH) (*p* = 0.001), and urine α-CTx (*p* < 0.001) of BC patients were found to be higher than the standard levels. After the BC patients were divided into subgroups on the basis of the presence of metastasis, the urine α-CTx levels (*p* = 0.001) were seen to be at critically high levels in those patients suffering from BC with metastasis. Though the BMD values in the lumbar spine (*p* < 0.001) and femoral neck (*p* = 0.001) were found to be significantly low in BC patients, no statistically substantial difference in the BMD levels of BC patients suffering from metastasis was observed. It was observed that urine α-CTx (specificity: 70%; sensitivity: 85%) values are critical factors that differentiate BC patients with metastasis from BC patients without metastasis. *Conclusions*: We found that alterations in bone turnover could be detected by using the values of urine α-CTx while differentiating BC patients with metastasis from BC patients without metastasis. Using the biochemical markers of bone turnover and BMD together would be pertinent for determining the level of metastasis present and examining the efficiency of bone density preservation therapy. Ideally, BMD measurement would be evaluated together with biochemical markers.

## 1. Introduction

The bones are the most common site of metastasis in several cancers, including breast, prostate, and thyroid. Early metastasis discovery is therefore as important as the early diagnosis of the elementary disease in terms of selecting the most appropriate treatment method and ensuring its success. Skeletal systems are often found to have metastasis, and this can often lead to life-threatening complexities and increased morbidity. As with patients with other forms of cancers, instances of bone metastasis increase the rates of mortality and morbidity in breast cancer (BC) patients [1].

Bone scintigraphy, along with laboratory methods and radiological tests, is commonly used to detect bone metastases, even though many limitations impact the specificity and sensitivity of these tests. Bone scintigraphy plays an important role in the detection of bone metastases [2]. In fact, no significant difference has been found between bone scintigraphy, positron emission tomography, and computed tomography (PET–CT) in terms of the effective detection of bone metastases in BC patients. The gold standard advocates for whole-body scintigraphy, but these techniques are characterized by high sensitivity and low specificity [2,3,4,5]. A positive correlation between bone mineral density (BMD) and BC risk in postmenopausal women was reported by a meta-analytical study, though its results were dependent on reports that varied in terms age, study design, measurement methods, and sites [6]. The loss of bone density and bone mass can be best marked by densitometry. In order to measure BMD, continuous measurements over a period of time are needed to highlight the decline in bone loss, whereas alterations in bone metabolism can be examined much earlier by using biochemical markers [7].

Radiological methods contribute to the early-stage diagnosis of bone metastasis, though only to a limited extent. Using biochemical markers for detecting metastatic bone disease, however, is comparatively inexpensive, reliable, and non-invasive, and multiple biochemical markers have actually been developed while detecting bone metastasis in patients. The potential oncological benefits of using such markers in detecting metastasis in bones have begun to be recognized in the last decade [8]. Their usage, at the appropriate times and under the appropriate conditions, makes it easier to detect alterations in the bone due to metastatic bone disease [9].

The aim of this study was to determine the power of bone turnover markers like urine CTx and NTx in detecting bone metastasis (bone-only) in breast cancer patients and to examine the possible relationship between BMD and parameters whose levels change as a result of bone metastasis.

## 2. Materials and Methods

The protocol for sample collection was approved by the Cerrahpasa Medical Faculty’s Ethical Committee (Date: 7 August 2020, No: 101831). The study was performed in accordance with the Helsinki Declaration, and informed consent was obtained from all patients and controls prior to their inclusion in the study.

This study was developed at the Department of Internal Medicine and Division of Oncology, along with the Department of Surgery, including Cerrahpasa Medical Faculty and Istanbul University-Cerrahpasa. It was consecutively performed on 30 postmenopausal BC patients suffering from bone metastases, 20 postmenopausal BC patients without bone metastases, and 20 age-matched healthy postmenopausal female controls.

All the subjects of this study were from Turkey; pregnant women, smokers, renal, hepatic, endocrine, or rheumatic patients, and individuals accustomed to chronic alcohol consumption were all excluded from this study. Multiple clinicopathological attributes, including the size, stage, and grade of the tumor; the number of participating axillary lymph nodes; histology; the status of the menopausal stage; and the status of estrogen receptors (ER) and progesterone receptors (PR) were examined according to the standards of the American Joint Committee for Cancer’s staging system [8].

### 2.1. Measurement of Bone Mineral Density (BMD)

BMD measurements were taken from the spine at L1–L2 and from the proximal femur bones with the aid of dual energy X-ray absorptiometry (DEXA) (Hologic, QDR 4000 densitometer, Bedford, MA, USA). Blood was drawn in the morning after patients had fasted for 12–14 h. Serum was obtained, after at least 30 min of clotting, via centrifuge at 2500× *g* for 15 min, after which some was removed in order to be used for measuring biochemical parameters and tumor markers. The remaining serum was preserved at −80 °C for analyzing and determining all other parameters; all hemolytic blood samples were rejected. The entire range of parameters for the samples was determined in a sole batch, wherein patient samples and control groups were studied together after completing the protocol obligations.

Calcium, phosphorous, parathyroid hormone (PTH), and all other biochemical criteria were determined using an automatic clinical analyzer (Roche COBAS Integra 800; Roche Diagnostics Corporation, Mannheim, Germany).

### 2.2. Measurement of Concentrations of Bone Turnover Markers 

#### 2.2.1. Measurement of Concentrations of Serum NTx 

The NTx levels in the serum were determined using commercial ELISA kits for humans (Serum CrossLaps ELISA, Osteometer, Denmark) in line with the manufacturer’s instructions. The inter-assay variation coefficient was found to be 6.2% (*n* = 15), while the intra-assay variation was 5.6% (*n* = 15).

#### 2.2.2. Measurement of Concentrations of Urine α-CTx

The NTx levels of urine α-CTx were determined using commercial ELISA kits for humans (Urine CrossLaps ELISA, Osteometer, Danimarka) in line with the manufacturer’s instructions. The inter-assay variation coefficient was found to be 6.2% (*n* = 15), while intra-assay variation was 5.6% (*n* = 15).

#### 2.2.3. Measurement of Concentrations of Tumor Markers

IMMULITE 2000 (DPC, Los Angeles, CA, USA) was used to measure tumor markers (CEA, CA 125, CA15-3). The chemiluminescent immunometric assay was utilized for the analysis of CA15-3 and the measurements of CEA and CA19-9.

## 3. Statistical Analysis

During the evaluation phase, SPSS (Statistical Package for Social Sciences) (SPSS for Windows 11.5; SPSS, Chicago, IL, USA) software was used to perform statistical analysis. The student’s *t*-test technique was employed for comparative analysis, and Pearson’s correlation test assisted in evaluating the correlation between partnering variables. ROC analysis determined the usability of various parameters as biomarkers; *p* < 0.05 was significant for deriving the sensitivity and specificity values.

## 4. Results

There was no significant difference in tumor marker concentrations across all age-matched breast cancer patients and the controls. In terms of the routine biochemical parameters, Ca (*p* = 0.001), PTH (*p* = 0.001), serum creatinine (*p* = 0.012), and serum urea (*p* = 0.012) levels were found to be high in BC patients, while urine creatinine (*p* = 0.003) and serum uric acid (*p* < 0.001) levels were found to be significantly low. The BMD analysis of the femoral neck BMD (*p* = 0.001) and lumbar spine BMD (*p* < 0.001) values showed significant decreases in individuals suffering from BC. Bone turnover markers (urine α-CTx, *p* < 0.001; serum NTx, *p* = 0.004) were found to be significantly higher in BC patients (Table 1). The absence of such a difference in tumor markers, as well as the significant differences between BMD and bone turnover marker levels, suggests that they can be used as useful markers for BC. It is important to note, however, that this difference may have been due to the presence of patients with bone metastases in the BC patient group.

Urine α-CTx levels in terms of both concentration (mg/mL) and concentration per amount of creatinine (μg/mmol Cr)) were found to be significantly higher in BC patients suffering from metastases compared to BC patients who were not. However, there were no differences in serum NTx levels, another bone turnover marker (BMD), or routine parameters (Table 2). This may indicate that urine α-CTx could be particularly useful in detecting bone metastasis in BC patients.

Correlation analysis results showed a powerful positive correlation between α-CTx and urine creatinine levels in all breast cancer patients (r = 0.880, *p* < 0.001; Figure 1). In terms of subgroups formed according to the presence of bone metastasis, α-CTx levels were found to be positively correlated with urine creatinine levels in both the group with bone metastasis (r = 0.959; *p* < 0.001) and the group without bone metastasis (r = 0.738; *p* < 0.001), though the strength of this correlation was found to be higher in the group with bone metastases. In addition, a positive correlation between tumor markers CA 125 and CA 15-3 (r = 0.809; *p* < 0.001) and a negative correlation between CA 125 and P (r = −0.559; *p* = 0.010; see Figure 2) were found in the group without bone metastasis.

The cut-off, sensitivity, and specificity values were derived to determine whether the biochemical parameters found through ROC analysis could be utilized as biomarkers for distinguishing BC patients from control groups and distinguishing BC patients with metastases from those without (Figure 3 and Figure 4). The parameters of Ca (AUC: 0.73, *p* = 0.042; sensitivity: 68%, specificity: 80%; cut-off point: 9.55 mg/dL), serum urea (AUC: 0.76, *p* = 0.022; sensitivity: 82%, specificity: 70% for a cut-off point of 26.50 mg/dL), and urine α-CTx (AUC: 0.79, *p* = 0.009; and sensitivity: 73%, specificity: 90% for a cut-off point of 4.30 μg/mmol Cr) were found to have a significant ability to differentiate BC patients from control groups (Table 3). The sole significant parameter that distinguished BC patients with metastases from BC patients without metastases was identified as urine α-CTx values (AUC: 0.83, *p* < 0.001, sensitivity: 85%, specificity: 70% when a cut-off point of 2.48 mg/mL was taken for urine α-CTx (mg/mL); AUC: 0.91, *p* < 0.001, sensitivity: 69%, specificity: 100% when a cut-off of 7.09 μg/mmol Cr was taken for urine α-CTx (μg/mmol Cr)) (Table 4). All of the data showed that the urine α-CTx (μg/mmol Cr) parameter could be used to distinguish breast cancer patients from healthy people, as well as that it is also a more specific biomarker for detecting breast cancer patients with bone metastases.

## 5. Discussion

The bones, lungs, liver, soft tissues, and brain are recognized as the most common sites of BC metastases. Considering the fact that there are no chances of cure for BC patients with metastasis, early diagnosis is crucially important. Biochemical markers quickly expose alterations in bone metabolism [7]. In one study, the values of serum NTx and urine α-CTx were found to be at higher levels in BC patients compared to control groups, while urine α-CTx values were recorded at considerably higher levels in BC with metastases. High levels of specificity and sensitivity were observed in the urine α-CTx values (sensitivity: 85%, specificity: 70% for urine α-CTx (mg/mL); sensitivity: 85%, specificity: 70% for urine α-CTx (μg/mmol Cr)) while differentiating BC patients with metastases from the BC patients without metastases. BMD values from the lumbar spine and femur bone were recorded at considerably lower levels in BC patients compared to the control groups, but no difference was observed based on the presence of metastases. These findings suggest that the most usable parameter in detecting the presence of metastases is urine α-CTx.

Multiple well-known factors tend to increase the threat of BC dissemination. Several risk factors related to bone-only metastasis have been investigated, and contradictory results have been found. In terms of the heterogeneous populations considered for this study, BC patients with elevated levels of urine NTx values during their latest tests are at twice as much risk for disease progression and at double or triple risk for skeletal damage (SREs: pathologic fracture, spinal cord compression, hypercalcemia of malignancy, and bone radiotherapy or surgery) compared to the patients with lower NTx levels [9]. Contemporary NTx levels were observed to have prognostic importance in the form of time-dependent variables. On the other hand, bone alkaline phosphatase (B-ALP) levels did not act as strong prognostic indicators. A random study on BC patients with metastasis confirmed the role of NTx levels in furnishing crucial prognostic information [10]. The median time for disease progression and overall median survival time were found to be shorter in patients with increased baseline NTx levels compared to patients with normal baseline NTx levels. Another similar study on BC patients with metastasis who were suffering from bone pain even after being treated with clodronate or pamidronate expressed that zoledronic acid tended to reduce their bone pain such that the improvements corresponded to a downward trend of urine NTx levels [11]. Additionally, an extensive study on BC patients with metastatic disease showed the prognostic value of NTx levels for the progression of bone disease, but did not show the same results for extraskeletal disease [12]. BC patients tended to display higher levels of serum NTx values in the healthy control groups of this study. Increased serum NTx levels have shown remarkable negative predictive implications in terms of the development of bone lesions and the potential continuity of the disease in BC patients.

CTx levels tend to be sensitive to the development of metastatic bone disease in BC patients. CTx marks the presence of bone resorption, and increased bone resorption is favorable for the progression of cancer cells. Evidence has suggested supplementary treatment with bisphosphonate is helpful in improving bone density, as well as decreasing bone metastasis and improving overall survival rates in postmenopausal women suffering from BC. Additional studies on AZURE have confirmed the strong predictive capability of bone turnover markers, namely CTx, P1NP, and 1-CTP, for bone-related recurrence [13]. None of these indicators have played a prognostic role in terms of either overall distant recurrence or the benefits of treating patients with zoledronic acid. All BC patients, including hypercalcemic patients with bone metastases (HC+) and normocalcemic patients with bone metastases (NC+), exhibited elevated levels of CTx isoforms compared to NC- patients or healthy postmenopausal control groups. The αL form, symbolizing the degeneration of newly-built bone, was found to be at an elevated level compared to the equivalent age-related isoforms [14]. BC frequently metastasizes to the bone, and studies have shown that 60–75% of primary breast cancers involve the diagnosis of bone metastases [15].

When the group of BC patients considered for this study was segregated into subgroups on the basis of the existence of metastases, the urine α-CTx values (*p* = 0.001) for both urine α-CTx (μg/mL) and urine α-CTx (μg/mmol Cr) were found to be substantially higher in BC patients suffering from metastases. Urine α-CTx values (sensitivity: 85%; specificity: 70%) were shown to be the sole parameter that differentiated BC patients with metastases from BC patients without metastases in this study. According to the data, the α-CTx levels are greatly affected by changes in bone turnover based on the metastatic encroachment of bone. Bone resorption causes the breakdown of collagen, which results in the leakage of biochemical markers into the circulatory system. Likewise, Leeming et al. [16] expressed that the estimated corresponding increases based on with the existence of one, two, or three metastases are 38%, 57%, and 81%, respectively. Considering the 17% intra-individual variation of the analysis, α-CTx acted as a sensitive biochemical indicator in terms of closely monitoring the cancer patients with the aim of detecting metastasis early. The degradation derivatives of α-CTx facilitate a specifically sensitive index of bone traction [17,18]. Earlier studies have also suggested that CTx markers can be utilized in evaluating skeletal metastases in patients suffering from breast cancer [19,20].

Early diagnosis and treatment are crucial for patients with metastases, though it is important to note that the core aims of such treatments are symptom reduction, incremental increases in quality of life, and possible life prolongation. The advancement of clinical and cellular studies and animal models will lead to an improved understanding of the critical invasion-metastasis cascade. Nonetheless, an extensive knowledge of bone metabolism-related risk factors would direct the formation of algorithms to detect the potential risk of bone disease in each patient, which, in turn, would trigger the engagement of special therapies to lead to bone disease-free patients. The results of this study revealed that no correlation exists between biochemical indicators of bone turnover and BMD outcomes, while biochemical markers related to bone metabolism offer valuable predictive information for patients with bone metastases [21,22,23,24].

A densitometer is the best tool for displaying bone mass and reduction in bone density with the passage of time. Our findings showed that changes in bone turnover can be detected with the help of urine α-CTx values, which also differentiate BC patients with metastasis from those without it. It is advisable to jointly utilize BMD and the biochemical markers of bone turnover for examining the extent of metastasis, as well as monitoring the efficacy of therapies used for bone preservation. Methods for detecting bone metastasis at an early stage should be implemented prior to the development of any complications.

## 6. Conclusions

Our results reveal the parallelism of fast bone turnover with low bone mass. BMD measurement provides information about the current state of bone mass, while biochemical indicators of bone turnover provide information about the rate of bone turnover and therefore potential future bone mass. Urinary α-CTx values can be used to detect changes in bone turnover in distinguishing BC patients with metastasis from BC patients without metastasis. It would be appropriate to use BMD and biochemical bone turnover markers together when evaluating the degree of metastasis and monitoring the effectiveness of bone-density-preserving therapy. Techniques used to detect bone metastasis in the early stages should be used before any complications develop. As a result, biochemical markers of bone densitometry and bone turnover are not substitutes for each other, but they can be complementary to each other. Ideally, BMD measurement is evaluated together with biochemical markers.

## Figures and Tables

**Figure 1 medicina-57-00880-f001:**
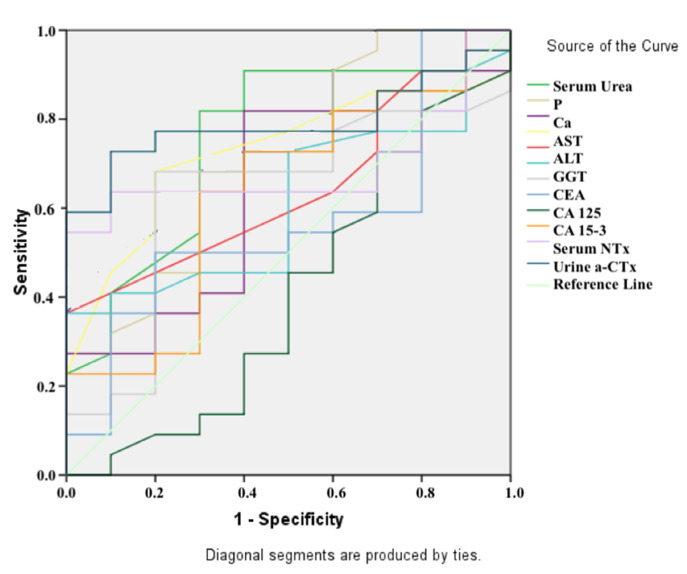
ROC analysis results to distinguish breast cancer patients from control individuals.

**Figure 2 medicina-57-00880-f002:**
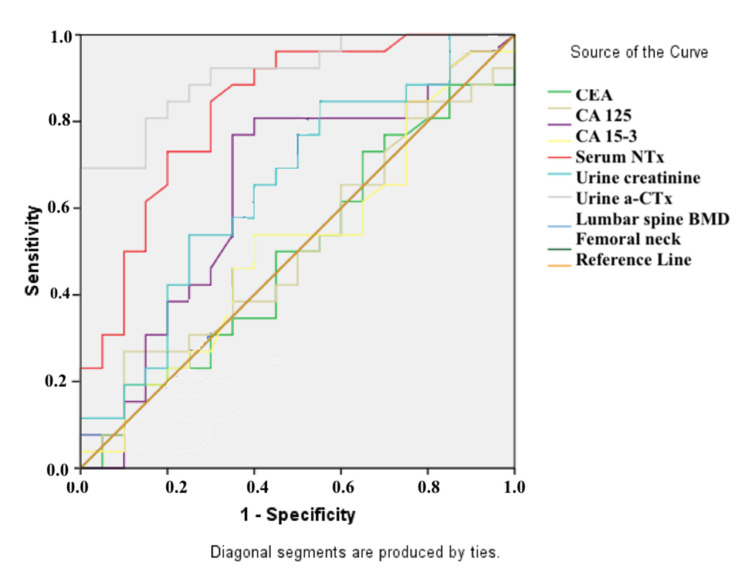
ROC analysis results for distinguishing breast cancer patients with metastasis from breast cancer patients without metastasis.

**Figure 3 medicina-57-00880-f003:**
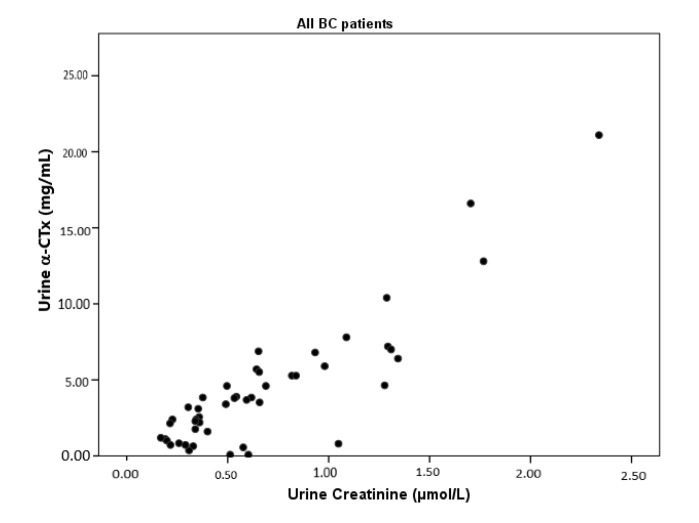
Correlation between α-CTx levels and urine creatinine for all breast cancer patients.

**Figure 4 medicina-57-00880-f004:**
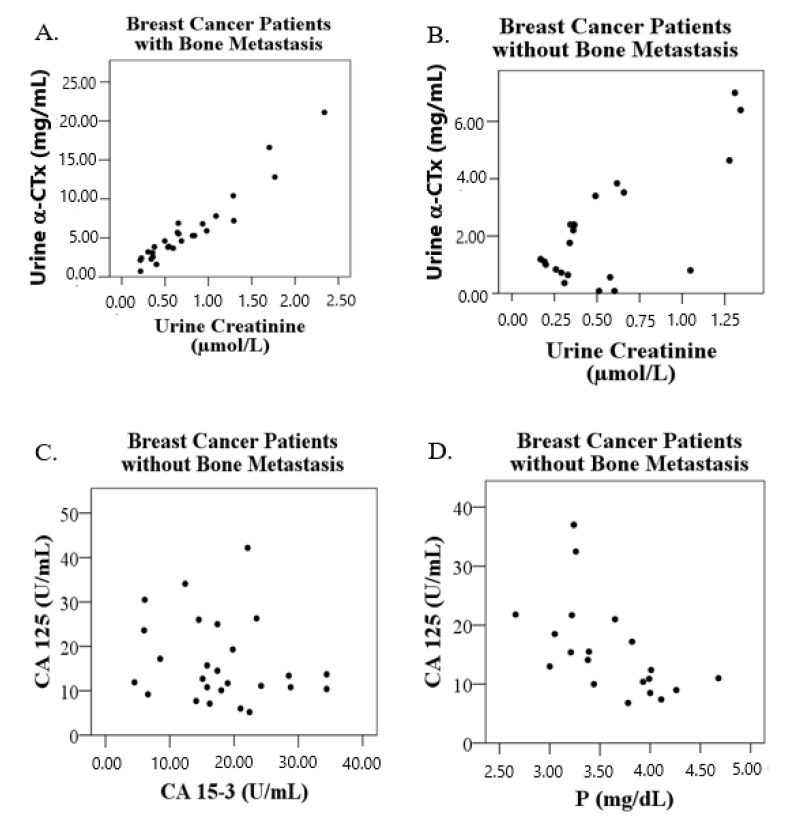
Correlation graphs of the bone metastasis status of patients with breast cancer. A. Correlation between urine α-CTx and urinary creatinine in breast cancer patients with bone metastases; B. Correlation between urine α-CTx and urinary creatinine in breast cancer patients without bone metastases; C. Correlation between CA-125 and CA 15-3 in breast cancer patients without bone metastases; D. Correlation between CA-125 and P in breast cancer patients without bone metastases.

**Table 1 medicina-57-00880-t001:** Biochemical characteristics of control and breast cancer patients.

	Control (*n* = 20)	Breast Cancer (*n* = 50)	*p*
	Mean ± S.D.	Mean ± S.D.	
**Age (years)**	55.55 ± 5.85	57.74 ± 10.28	0.375
**CEA (ng/mL)**	1.61 ± 1.55	2.48 ± 2.68	0.179
**CA 125 (U/mL)**	16.96 ± 10.63	16.04 ± 8.63	0.711
**CA 15-3 (U/mL)**	13.06 ± 6.50	18.50 ± 16.24	0.160
**AST (U/L)**	18.53 ± 4.02	21.53 ± 8.38	0.143
**ALT (U/L)**	17.53 ± 7.62	21.67 ± 10.33	0.123
**GGT (U/L)**	23.75 ± 17.21	50.13 ± 104.29	0.420
**ALP (U/L)**	60.53 ± 20.08	183.97 ± 623.50	0.408
**PTH (pg/mL)**	57.80 ± 11.95	71.85 ± 19.18	**0.001**
**P (mg/dL)**	3.24 ± 0.47	3.46 ± 0.57	0.137
**Ca (mg/dL)**	10.07 ± 1.31	9.37 ± 0.33	**0.001**
**Creatinine (mg/dL)**	0.76 ± 0.10	0.86 ± 0.15	**0.012**
**Urea (mg/dL)**	0.76 ± 0.10	0.86 ± 0.15	**0.012**
**Uric acid (mg/dL)**	0.80 ± 0.00	0.90 ± 0.10	**0.000**
**Serum NTx (nMBCE/L)**	7.70 ± 2.02	13.31 ± 7.53	**0.004**
**Urine α-CTx (μg/mmol Cr)**	2.86 ± 1.67	6.12 ± 2.84	**0.000**
**Lumbar spine BMD (g/cm^2^)**	0.90 ± 0.10	0.80 ± 0.00	**0.000**
**Femoral neck BMD (g/cm^2^)**	0.73 ± 0.15	0.60 ± 0.00	**0.001**

Bolded *p* values indicate statistical significance.

**Table 2 medicina-57-00880-t002:** Biochemical parameters in subgroups divided according to the presence of metastasis in breast cancer patients.

	Breast Cancer without Metastasis *n* = 20	Breast Cancer with Metastasis *n* = 30	*p*
	Mean ± S.D.	Mean ± S.D.	
**Age (years)**	55.30 ± 11.55	59.56 ± 9.02	0.163
**CEA (ng/mL)**	2.89 ± 3.80	2.17 ± 1.38	0.367
**CA 125 (U/mL)**	15.70 ± 8.00	16.28 ± 9.20	0.824
**CA 15-3 (U/mL)**	19.23 ± 23.16	17.74 ± 8.04	0.757
**AST (U/L)**	23.33 ± 12.05	20.24 ± 4.00	0.307
**ALT (U/L)**	23.33 ± 12.78	20.42 ± 8.10	0.372
**GGT (U/L)**	75.63 ± 157.19	31.59 ± 25.63	0.284
**ALP (U/L)**	78.55 ± 34.66	230.41 ± 758.64	0.377
**PTH (pg/mL)**	69.40 ± 19.84	73.67 ± 18.85	0.457
**P (mg/dL)**	3.60 ± 0.50	3.35 ± 0.60	0.137
**Ca (mg/dL)**	9.45.13.60	10.63 ± 1.21	0.103
**Creatinine (mg/dL)**	0.84 ± 0.14	0.88 ± 0.16	0.421
**Urea (mg/dL)**	0.84 ± 0.14	0.88 ± 0.16	0.421
**Uric acid (mg/dL)**	0.80 ± 0.00	0.80 ± 0.00	1.000
**Serum NTx (nMBCE/L)**	13.36 ± 7.38	15.17 ± 11.32	0.536
**Urine α-CTx (mg/mL)**	2.13 ± 2.05	5.91 ± 4.61	**0.001**
**Urine α-CTx (μg/mmol Cr)**	3.96 ± 2.41	7.33 ± 2.34	**0.001**
**Lumbar spine BMD (g/cm^2^)**	0.80 ± 0.00	0.80 ± 0.00	1.000
**Femoral neck BMD (g/cm^2^)**	0.60 ± 0.00	0.60 ± 0.00	1.000

Bolded *p* values indicate statistical significance.

**Table 3 medicina-57-00880-t003:** Cut-off points, sensitivity, specificity, and AUC to distinguish breast cancer patients from control individuals.

Variables	AUC	*p*	Asymptotic 95% CI		Cut-Off Points	Sensitivity	Specificity
			Lower Bound	Upper Bound			
**Serum Urea** **(mg/dL)**	0.76	**0.022**	0.57	0.94	**26.50**	**82%**	**70%**
**ALP** **(U/L)**	0.71	0.056	0.52	0.91	N.S.	N.S.	N.S.
**P** **(mg/dL)**	0.64	0.208	0.43	0.85	N.S.	N.S.	N.S.
**Ca** **(mg/dL)**	0.73	**0.042**	0.55	0.90	**9.55**	**68%**	**80%**
**AST** **(U/L)**	0.65	0.193	0.45	0.84	N.S.	N.S.	N.S.
**ALT** **(U/L)**	0.61	0.329	0.41	0.81	N.S.	N.S.	N.S.
**GGT** **(U/L)**	0.63	0.238	0.42	0.84	N.S.	N.S.	N.S.
**CEA** **(ng/mL)**	0.57	0.542	0.35	0.78	N.S.	N.S.	N.S.
**CA 125** **(U/mL)**	0.40	0.393	0.18	0.63	N.S.	N.S.	N.S.
**CA 15-3** **(U/mL)**	0.63	0.238	0.42	0.84	N.S.	N.S.	N.S.
**Serum NTx (nMBCE/L)**	0.69	0.088	0.51	0.87	N.S.	N.S.	N.S.
**Urine α-CTx (μg/mmol Cr)**	0.79	**0.009**	0.63	0.95	**4.30**	**73%**	**90%**

N.S. = non-significant; bold indicates statistical significance.

**Table 4 medicina-57-00880-t004:** Cut-off points, sensitivity, specificity, and AUC for distinguishing breast cancer patients with metastasis from breast cancer patients without metastasis.

Variables	AUC	*p*	Asymptotic 95% CI		Cut-Off Values	Sensitivity	Specificity
			Lower Bound	Upper Bound			
**CEA** **(ng/mL)**	0.49	0.921	0.32	0.66	N.S.	N.S.	N.S.
**oCA 125** **(U/mL)**	0.51	0.929	0.34	0.68	N.S.	N.S.	N.S.
**CA 15-3** **(U/mL)**	0.64	0.106	0.47	0.81	N.S.	N.S.	N.S.
**Serum NTx** **(nMBCE/L)**	0.52	0.859	0.34	0.69	N.S.	N.S.	N.S.
**Urine α-CTx** **(mg/mL)**	0.83	**0.000**	0.71	0.95	**2.48**	**85%**	**70%**
**Urine Creatinine** **(µmol/L)**	0.65	0.084	0.49	0.81	N.S.	N.S.	N.S.
**Urine α-CTx** **(μg/mmol Cr)**	0.91	**0.000**	0.83	0.99	**7.09**	**69%**	**100%**
**Lumbar spine BMD** **(g/cm^2^)**	0.50	1.000	0.33	0.67	N.S.	N.S.	N.S.
**Femoral neck BMD** **(g/cm^2^)**	0.50	1.000	0.33	0.67	N.S.	N.S.	N.S.

N.S = non-significant; bold indicates statistical significance.

## Data Availability

The data supporting the revelations of this study can be retrieved from the corresponding author upon reasonable request.

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
