# Peer review of "Association between Bone Mineral Density and Bone Turnover Markers in Breast Cancer Patients and Bone-Only Metastasis"

_medicina, 2021, doi:10.3390/medicina57090880_

Round 1

Reviewer 1 Report

The paper has been changed to an acceptable edition for publication.

Reviewer 2 Report

The authors have answered all my concerns and the manuscript has been improved. I do not have any further comments.

Reviewer 3 Report

Thank you for your great manuscript about "Association between bone mineral density and bone turnover markers in breast cancer patients and bone-only metastasis"

This manuscript is helpful for detection bone metastasis in Breast cancer patients.

If possible, In Table 1, it would be easier to see if a value with statistical significance is marked with an "*" in the column P.

This manuscript is a resubmission of an earlier submission. The following is a list of the peer review reports and author responses from that submission.

Round 1

Reviewer 1 Report

By analyzing the bone turnover markers from breast cancer patients and age matched healthy women, Berrin Papila Kundaktepe and colleagues found that urinary α-CTx can be used as an independent factor to predict the bone metastasis of breast cancer. It is an interesting study with clinical value, most of the experiments are clear and well performed. However, there are still some major flaws need to be explained or supplemented.

  1. In line 26, is “BC” short for “Bone cancer” or “Breast cancer”? This makes me very confused; I think in the rest of the manuscript, “BC” stands for “Breast cancer”, but not “Bone cancer”. Please correct or clarify it.
  2. In abstract, the section of “Methods” should include the information of bone turn over marker detection.
  3. All abbreviations should be explained the first time they appear in your article. For example: PTH in line 34; HC and NC in line 216.
  4. The “INTRODUCTION” needs to be rewrite with more logic, as among different paragraphs, the authors just repeated the same information using different expressions. The background of the first sentence “The highest number of deaths is caused by coronary artery disease; however, cancer causes the second-highest number of deaths” is too big, but too far away from this project.
  5. Please re-organize the result part, do not simply list lots of data but summarize and interpret their meaning to the readers.
  6. From line 148-157, the authors performed the correlation analysis, where are the figures? Please include them.
  7. The authors need to rewrite the “DISCUSSION” part. Put your story in the big picture, what your findings contribute to this field, not just provide another “INTRODUCTION”.

Reviewer 2 Report

1. The whole paper needs serious English editing; some sentences are not understandable.

2. the title must be rewritten.

3. the introduction is not clear; should be rewritten.

4. The study does not have novelty.